# Factors associated with unmet need for family planning in sub-Saharan Africa: A multilevel multinomial logistic regression analysis

**Achamyeleh Birhanu Teshale** [ID] *

Department of Epidemiology and Biostatistics, Institute of Public Health, College of Medicine and Health Sciences, University of Gondar, Gondar, Ethiopia

* achambir08@gmail.com

**Data Availability Statement:** All relevant data are within the manuscript.

**Funding:** The author(s) received no specific funding for this work.

## Abstract

### Background

More than one out of every ten married women in the world, and one out of every five women in Africa, have unmet family planning needs. Despite this, studies concerning sub-Saharan Africa as well as the community-level factors that may influence the unmet need for family planning are scarce.

### Objective

To assess factors associated with unmet need for family planning in sub-Saharan Africa.

### Methods

This study used the nineteen demographic and health surveys (DHS) conducted between 2015 and 2020 in sub-Saharan Africa. A total weighted sample of 175, 820 women of reproductive age who were married during the survey was used for this study. To assess the factors associated with unmet need for family planning, I have employed a multilevel multinomial logistic regression model. After selecting variables using the bivariable analysis, a multivariable model was fitted. Finally, an adjusted relative risk ratio with its 95% Confidence Interval was reported and variables with a p-value less than 0.05 were declared to be significant predictors of unmet need for family planning.

### Result

The overall prevalence of unmet need for family planning in sub-Saharan Africa was 23.70%, of which unmet need for spacing and limiting was 15.81% and 7.90% respectively. In the multivariable multilevel multinomial model, women's age, women's education, age at cohabitation, heard about family planning through media, parity, number of under-five children, and knowledge about modern contraceptive methods were among the individual-level factors that were associated with both the unmet need for spacing and limiting. Place of residence, community level of women illiteracy, and region were among the community-level factors that were associated with both unmet needs for spacing and limiting. Household size

**Competing interests:** The authors have declared that no competing interests exist.

**Abbreviations:** CI, Confidence Interval; DHS, Demographic and Health Surveys; ICC, Intraclass Correlation Coefficient; PCV, Proportional Change in Variance; RRR, Relative Risk Ratio.

and visiting the health facility in the last 12 months were associated with unmet need for spacing only and husband education was associated with unmet need for limiting only.

## Conclusion

Unmet need for family planning in sub-Saharan Africa was high. Both individual and community level factors were associated with both unmet need for spacing and limiting. Therefore, it is better to consider interventions at both individual and community levels.

## Background

Even though reproductive and sexual health care services are very crucial, their availability to women of reproductive age is quite limited [1]. One of these services is family planning and a lack of family planning or not using it regularly or appropriately leads to unwanted pregnancy, which is one of the leading causes of maternal mortality [2]. Unintended pregnancy affects almost every one of life and is a huge and global public health issue. It is linked to parental stress, induced abortion, infertility, negative physical, mental, social, and economic consequences, as well as maternal and child mortality [1,3,4].

The most common cause of unwanted pregnancies among married women is an unmet demand for family planning [5,6]. It is the percentage of women who are currently married or in a partnership who decide to stop or delay pregnancy/childbearing but without using contraception [7–9].

In 2017, the majority (63%) of reproductive-age women who were married or in a partnership used contraception globally; however, there was significant variation between regions. More than one out of every ten married women in the world, and one out of every five women in Africa, have unmet family planning needs [10].

According to different pieces of literature, individual-level factors such as women age, women education, husband education, wealth index, media exposure, age at first marriage, number of living children, parity, household size, health care decision making, knowledge about family planning method and different community-level factors are related with such higher occurrence of unmet need for family planning [11–24].

Unmet family planning needs are an important topic to consider when lobbying for reproductive health, developing family planning policy, and monitoring and assessing family planning programs. Despite the fact, prior studies have addressed the unmet need for family planning, most of them focus on a single country, and many of them ignore community-level factors that may influence the unmet need for family planning. Therefore, this study aimed to assess factors associated with unmet need for family planning among married women of reproductive age in sub-Saharan Africa using a multilevel multinomial logistic regression analysis.

## Methods

### Data source and study population

This study used the nineteen demographic and health surveys (DHS) conducted between 2015 and 2020 in sub-Saharan Africa. The DHS is a survey conducted every five years. It is based on a two-stage sampling technique. In the first stage, clusters/enumeration areas were selected using each country's most recent population and housing census as a sampling frame, and in the second stage; households were sampled from the newly created household listing. The

**Table 1. Sample size in each country, study year, and total sample size.**

| Country | Year | Total population (Weighted) | Frequency |
|---|---|---|---|
| Angola | 2015 | 7,957 | 4.53 |
| Benin | 2017/18 | 11,168 | 6.35 |
| Burundi | 2016/17 | 9,782 | 5.56 |
| Cameroon | 2018/19 | 7,748 | 4.41 |
| Ethiopia | 2016 | 10,223 | 5.81 |
| Gambia | 2019/20 | 7,526 | 4.28 |
| Guinea | 2018 | 7,727 | 4.39 |
| Liberia | 2019/20 | 4,216 | 2.40 |
| Mali | 2018 | 8,567 | 4.87 |
| Nigeria | 2018 | 29,090 | 16.55 |
| Rwanda | 2015 | 6,978 | 3.97 |
| Sierra Leone | 2019 | 9,715 | 5.53 |
| Senegal | 2019 | 5,657 | 3.22 |
| Chad | 2015 | 13,184 | 7.50 |
| Tanzania | 2015/16 | 8,210 | 4.67 |
| Uganda | 2016 | 11,223 | 6.38 |
| South Africa | 2016 | 3,050 | 1.73 |
| Zambia | 2018/19 | 7,648 | 4.35 |
| Zimbabwe | 2015 | 6,151 | 3.50 |
| Total | 2015–2020 | 175,820 | 100.00 |

survey target groups were women and men of reproductive age in randomly selected households of each country. The detailed information was collected on background characteristics, maternal and child health, HIV/AIDS, domestic violence, family planning services, and other important public health problems. For this study, we have appended women's data of each country. Only women of reproductive age who were married were included since they are exposed to regular sexual intercourse. Nineteen countries' DHS data with a total weighted sample of 175, 820 women of reproductive age who were married during the survey was used for this study (Table 1). Further information regarding the DHS methodology can be found elsewhere [25].

## Variables of the study

**Outcome variable.** Unmet need for family planning was the dependent variable. It had three categories; the unmet need for spacing (women who wanted to wait/delay in having another child, but not using any form of contraception), unmet need for limiting (women who did not want any more children, but did not use any contraceptive method), and no unmet need. This classification was made using the DHS's recently revised definition of unmet need for family planning [26].

**Explanatory variables.** Both individual and community level variables were incorporated as explanatory variables after reviewing the DHS data.

*Individual-level variables.* Women age, women and husbands education, wealth index, age at cohabitation, health care decision making, visited a health facility in the last 12 months, hearing about family planning methods by television, radio, or/and newspaper/magazines, parity, number of under-five children, household size, and having knowledge on family planning methods were the individual-level variables.

*Community-level variables.* The community-level variables incorporated for this study were; residence, African region, community level of women illiteracy, community level of

media exposure, community level of child care burden, and community poverty level. Except for residence and African region, all other community-level variables were created by aggregating individual-level variables into cluster/community level variables as follows:

**Community-level of women illiteracy;** the proportion of women with no formal education derived from data on women's level of education and categorized as low and high community level of women education (using national median value).

**Community-level childcare burden;** the proportion of women who had five or more under-five children derived from data on the total number of under-five children and categorized as low and high community level of childcare burden based on a national median value.

**Community poverty level;** the proportion of women in the poorest and poorer quintiles derived from data on wealth index and categorized as low and high poverty communities based on a national median value.

**Community-level media exposure;** was measured by the proportion of women who had been exposed to family planning methods through at least one media (either television, radio, and/or newspaper/magazine) and categorized as low and high community level of media exposure like the above community-level variables.

## Data management and statistical analysis

The data was appended, recoded, and analyzed using Stata version 16 software. To restore representativeness and to get an appropriate statistical estimate, every analysis was based on weighting. Both descriptive (frequencies and percentages) and analytical analysis (multilevel multinomial logistic regression) were done and reported.

**Model building.** I have employed a multilevel multinomial logistic regression analysis, which includes both random effect and fixed-effect models. This statistical approach was done in Stata version 16 software via a Generalized Structural Equation Modelling (with the logit link function) using the *gsem* Stata command. Four different nested models were fitted. The models were null model (containing only the outcome variable), model 1 (a model fitted using individual-level variables only), model 2 (a model fitted using community-level variables), and model 3 (fitted using both individual and community level variables). The final interpretation of results were based on the best model among them, which is selected through the use of the log-likelihood and Akaike's information criteria. Both bivariable (to select eligible variables for the multivariable analysis) and multivariable analysis was employed. P-value less than 0.20 was used to retain and include variables in the multivariable analysis. Finally, the adjusted Relative Risk Ratio (RRR) with 95% Confidence Interval (CI) was reported and variables with p value<0.05, in the multivariable analysis, were declared to be significant predictors of unmet need for family planning (unmet need for spacing and limiting). Furthermore, I have checked Multicollinearity between explanatory variables using Variance Inflation Factor (VIF) and found that there was no Multicollinearity (the VIF ranges from 1.03 to 2.09 with the mean VIF of 1.45)

In the random-effects analysis, to assess the variability of unmet need for family planning between clusters/communities, both Intraclass Correlation Coefficient (ICC) and Proportional change in Variance (PCV) were calculated.

## Ethical consideration

Since I have used the publicly available data, ethical approval was not required. However, I have accessed the data set from the DHS website (https://dhsprogram.com) after telling the purpose of the study.

## Results

### Sociodemographic characteristics of respondents

A total weighted sample of 175,820 women of reproductive age who were married was used for the final analysis. Most (16.55%) of the study participants were from Nigeria (Table 1). The median age of the study participants was 30 (IQR = 25–38) years. The majority (43.48%) had no formal education and only 4.64% had higher education. More than half (52.67%) of the respondents were married at the age of 18 and older. Around 58.94% of the respondents visit the health facility during the last 12 months. The majority of the respondents did not heard about family planning methods through radio (66.40%), television (83.54%), and newspaper/magazine (93.69%). More than half (54.57%) of the respondents were from a household size of five and above and 93.45% of the respondents know about modern family planning methods. Regarding the place of residence and region, 64.91% and 47.59% of the respondents were from rural residence and western African region respectively (Table 2).

### Unmet need for family planning in sub-Saharan Africa

The overall prevalence of unmet need for family planning was 23.70% (95%CI; 23.50, 23.90). The prevalence of unmet need for spacing and limiting was 15.81% (95%CI; 15.64, 15.98) and 7.90% (95%CI; 7.77, 8.02) respectively.

### Factors associated with unmet need for family planning in sub-Saharan Africa

A multilevel multinomial logistic regression analysis was fitted to assess the factors associated with unmet need for family planning. In this study, in the multivariable multilevel multinomial regression analysis, both individual level and community level variables were associated with both unmet needs for spacing and limiting.

**Factors associated with unmet need for spacing.** Women age, women education, age at cohabitation, health care decision making, visiting health facility in the last 12 months, heard about family planning through reading newspaper/magazine and watching television, parity, number of under-five children, household size, knowledge about modern contraceptive methods were among the individual-level factors that were associated with unmet need for spacing. Place of residence, community level of women illiteracy, and region were among the community-level factors that were associated with unmet need for spacing. Being in an older age group was associated with a lower risk of having an unmet need for spacing compared with younger-aged women. Women with primary education and higher education had 10% (RRR = 1.10, 95%CI; 1.05, 1.15) higher risk and 12% (RRR = 0.88, 95%CI; 0.78, 0.99) lower risks of having an unmet need for spacing, respectively. Women who have got married at or above the age of 18 had 32% (RRR = 1.32, 95%CI; 1.27, 1.37) higher risk of having an unmet need for spacing as compared to their counterparts. Regarding health care decision, a decision made by both women and her husband, husband only, and others had 6% (RRR = 0.96, 95% CI; 0.90, 0.98) lower risk, 5% (RRR = 0.95, 95%CI; 0.90, 0.99) lower risk, and 24% (RRR = 1.24, 95%CI; 1.04, 1.49) higher risk of having an unmet need for spacing as compared to decisions made by women only respectively. Being visiting the health facility in the last 12 months was associated with 7% (RRR = 1.07, 95%CI; 1.04, 1.11) higher risks of unmet need for spacing as compared to their counterparts. Being heard about family planning through television and newspaper/magazine were associated with 11% (both have similar figure) lower risks of having an unmet need for spacing respectively. Higher parity, a higher number of under-five children, being from the nuclear family, and knowing modern contraceptive methods were also

**Table 2. Sociodemographic characteristics of respondents and their partner.**

| Characteristics | Frequency (N = 175,820) | Percentage (%) |
|---|---|---|
| Maternal age | | |
| 15–19 | 11,807 | 6.27 |
| 20–24 | 29,073 | 16.54 |
| 25–29 | 37,945 | 21.58 |
| 30–34 | 32,973 | 18.75 |
| 35–39 | 28,373 | 16.14 |
| 40–44 | 19,989 | 11.37 |
| 45–49 | 15,660 | 8.91 |
| Wealth index | | |
| Poorest | 35,085 | 19.95 |
| Poorer | 36,153 | 20.56 |
| Middle | 34,900 | 19.85 |
| Rich | 35,059 | 19.94 |
| Richer | 34,623 | 19.69 |
| Women education | | |
| No formal education | 76,881 | 43.73 |
| Primary | 51,248 | 29.15 |
| Secondary | 39,536 | 22.49 |
| Higher | 8,155 | 4.64 |
| Husband education | | |
| No formal education | 69,155 | 39.33 |
| Primary | 45,951 | 26.14 |
| Secondary | 45,894 | 26.10 |
| Higher | 14,820 | 8.43 |
| Age at cohabitation (years) | | |
| <18 | 83,222 | 47.33 |
| ≥18 | 92,598 | 52.67 |
| Health care decision | | |
| Women only | 28,692 | 16.32 |
| Both | 69,493 | 39.53 |
| Husband only | 76,454 | 43.48 |
| Others* | 1,181 | 0.67 |
| Visited Health facility during the last 12 months | | |
| No | 72,184 | 41.06 |
| Yes | 103,636 | 58.94 |
| Heard about family planning through the radio last few months | | |
| No | 116,743 | 66.40 |
| Yes | 59,077 | 33.60 |
| Heard family planning in newspaper magazine last few months | | |
| No | 164,730 | 93.69 |
| Yes | 11,090 | 6.31 |
| Heard family planning on television last few months | | |
| No | 146,874 | 83.54 |
| Yes | 28,946 | 16.46 |
| Parity | | |
| 0 | 11,618 | 6.61 |
| 1 | 24,785 | 14.10 |
| 2–4 | 77,391 | 44.02 |
| 5 and above | 62,026 | 35.28 |
| Number of under 5 children | | |
| None | 36,805 | 20.93 |
| One | 58,546 | 33.30 |
| Two | 49,100 | 27.93 |
| Three and above | 31,369 | 17.84 |
| Household size | | |
| 1–2 | 9,662 | 5.50 |
| 3–4 | 70,206 | 39.93 |
| 5 and above | 95,952 | 54.57 |

(*Continued*)

**Table 2.** (Continued)

| Characteristics | Frequency (N = 175,820) | Percentage (%) |
|---|---|---|
| Having knowledge about modern family planning methods | | |
| No | 11,509 | 6.55 |
| Yes | 164,311 | 93.45 |
| Residence | | |
| Urban | 61,692 | 35.09 |
| Rural | 114,128 | 64.91 |
| Community-level of women illiteracy | | |
| Low | 88,955 | 50.59 |
| High | 86,865 | 49.41 |
| Community-level media exposure | | |
| Low | 87,389 | 49.70 |
| high | 88,431 | 50.30 |
| Community-level child care burden | | |
| Low | 87,781 | 49.93 |
| high | 88,039 | 50.07 |
| Community poverty level | | |
| Low | 91,437 | 52.01 |
| High | 84,383 | 47.99 |
| African region | | |
| Eastern | 60,215 | 34.25 |
| Western | 83,666 | 47.59 |
| Central | 28,889 | 16.43 |
| Southern | 3,050 | 1.73 |

Note

* = someone else and other individuals.

associated with higher risks of having an unmet need for spacing. Being from rural residence (RRR = 0.86, 95%CI; 0.82, 0.91) and communities with higher illiteracy level (RRR = 0.94, 95% CI; 0.90, 0.99) were associated with lower risks of having an unmet need for spacing as compared to their counterparts respectively. Moreover, being from the western African region (RRR = 1.25, 95%CI; 1.19, 1.33), central African region (RRR = 1.48, 95%CI; 1.39, 1.57), and southern African region (RRR = 0.71, 95%CI; 0.57, 0.89) were associated with having an unmet need for spacing (Table 3).

**Factors associated with unmet need for limiting.** Women age, women education, husband education, age at cohabitation, health care decision making, heard about family planning methods through magazine or newspaper, parity, number of under-five children, and knowledge about modern contraceptive methods were the individual-level factors that were associated with unmet need for limiting. While place of residence, community level of women illiteracy, and region were among the community-level factors that were associated with unmet need for limiting. Women in the age group 20–24 had 32% (RRR = 0.68, 95%CI; 0.52, 0.90) lower risks for having an unmet need for spacing. While women in the age group 30–34, 35–39, 40–44, and 45–49 had 1.51 (RRR = 1.51, 95%CI; 1.16, 1.96), 2.45 (RRR = 2.45, 95%CI; 1.87, 3.16), 3.27 (RRR = 3.27, 95%CI; 2.49, 4.28), and 2.43 (RRR = 2.43, 95%CI; 1.86, 3.18) times higher risk of having unmet need for limiting respectively. Having primary education is associated with a higher risk of having an unmet need for limiting and being having a higher education is associated with a lower risk of having an unmet need for limiting. Women with their husbands having primary education and secondary education had a higher risk of having an unmet need for limiting as compared to those women with their husbands not having a formal education. Women who have married at the age of 18 and above had a 1.06 (RRR = 1.06,

**Table 3. Multilevel multinomial logistic regression analysis in assessing the factors associated with unmet need for family planning for (both spacing and limiting).**

| Characteristics | Null model | Model 1 | | Model 2 | | Model 3 | |
|---|---|---|---|---|---|---|---|
| | | **Individual-level characteristics** | | **Community-level characteristics** | | **Both individual and community-level characteristics** | |
| | | Unmet need for spacing RRR (95%CI) | Unmet need for limiting RRR (95%CI) | Unmet need for spacing RRR (95%CI) | Unmet need for limiting RRR (95%CI) | Unmet need for spacing RRR (95%CI) | Unmet need for limiting RRR (95%CI) |
| Maternal age | | | | | | | |
| 15–19 | | 1.00 | 1.00 | | | 1.00 | 1.00 |
| 20–24 | | 0.65 (0.61, 0.70) | 0.69 (0.53, 0.91) | | | 0.66 (0.62, 0.71)*** | 0.68 (0.52, 0.90)** |
| 25–29 | | 0.50 (0.46, 0.54) | 0.87 (0.67, 1.13) | | | 0.51 (0.47, 0.55)*** | 0.85 (0.65, 1.10) |
| 30–34 | | 0.41 (0.38, 0.45) | 1.57 (1.2, 2.04) | | | 0.42 (0.39, 0.46)*** | 1.51 (1.16, 1.96)** |
| 35–39 | | 0.31 (0.28, 0.34) | 2.55 (1.96, 3.31) | | | 0.32 (0.29, 0.35)*** | 2.45 (1.87, 3.16)*** |
| 40–44 | | 0.18 (0.16, 0.20) | 3.43 (2.6, 4.49) | | | 0.18 (0.16, 0.20)*** | 3.27 (2.49, 4.28)*** |
| 45–49 | | 0.07 (0.06, 0.08) | 2.55 (1.95, 3.34) | | | 0.07 (0.06, 0.08)*** | 2.43 (1.86, 3.18)*** |
| Wealth index | | | | | | | |
| Poorest | | 1.00 | 1.00 | | | 1.00 | 1.00 |
| Poorer | | 1.03 (0.98, 1.08) | 1.05 (0.98, 1.13) | | | 1.01 (0.96, 1.06) | 1.05 (0.98, 1.12) |
| Middle | | 1.06 (1.01, 1.12) | 1.02 (0.95, 1.09) | | | 1.01 (0.96, 1.06) | 0.99 (0.93, 1.07) |
| Rich | | 1.06 (1.00, 1.13) | 1.07 (0.99, 1.15) | | | 0.98 (0.92, 1.04) | 1.01 (0.94, 1.09) |
| Richer | | 1.08 (1.01, 1.16) | 1.14 (1.04, 1.24) | | | 0.95 (0.88, 1.03) | 1.03 (0.94, 1.13) |
| Women education | | | | | | | |
| No formal education | | 1.00 | 1.00 | | | 1.00 | 1.00 |
| Primary | | 1.06 (1.02, 1.11) | 1.10 (1.04, 1.17) | | | 1.10 (1.05, 1.15)*** | 1.07 (1.00, 1.13)* |
| Secondary | | 0.98 (0.93, 1.04) | 1.05 (0.96, 1.15) | | | 0.97 (0.92, 1.02) | 0.98 (0.90, 1.08) |
| Higher | | 0.89 (0.79, 1.01) | 0.89 (0.75, 1.05) | | | 0.88 (0.78, 0.99)* | 0.84 (0.71, 0.99)* |
| Husband education | | | | | | | |
| No formal education | | 1.00 | 1.00 | | | 1.00 | 1.00 |
| Primary | | 0.93 (0.89, 0.98) | 1.11 (1.05, 1.18) | | | 0.99 (0.94, 1.040) | 1.06 (1.02, 1.16)* |
| Secondary | | 0.97 (0.92, 1.02) | 1.12 (1.04, 1.21) | | | 0.96 (0.91, 1.01) | 1.10 (1.01, 1.18)* |
| Higher | | 0.96 (0.88, 1.04) | 1.07 (0.95, 1.20) | | | 0.95 (0.87, 1.03) | 1.06 (0.94, 1.20) |
| Age at cohabitation (years) | | | | | | | |
| <18 | | 1.00 | 1.00 | | | 1.00 | 1.00 |
| ≥18 | | 1.30 (1.25, 1.35) | 1.07 (1.01, 1.12) | | | 1.32 (1.27, 1.37)*** | 1.06 (1.01, 1.11)* |
| Health care decision | | | | | | | |
| Women only | | 1.00 | 1.00 | | | 1.00 | 1.00 |
| Both | | 0.95 (0.90, 0.98) | 0.81 (0.76, 0.86) | | | 0.94 (0.90, 0.98)** | 0.82 (0.77, 0.87)*** |
| Husband only | | 1.01 (0.96, 1.06) | 0.68 (0.64, 0.73) | | | 0.95 (0.90, 0.99)* | 0.71 (0.67, 0.76)*** |
| Others | | 1.31 (1.09, 1.56) | 0.80 (0.58, 1.10) | | | 1.24 (1.04, 1.49)* | 0.83 (0.60, 1.14) |
| Visited Health facility during the last 12 months | | | | | | | |
| No | | 1.00 | 1.00 | | | 1.00 | 1.00 |
| Yes | | 1.04 (1.01, 1.08) | 1.01 (0.96, 1.06) | | | 1.07 (1.04, 1.11)*** | 0.99 (0.94, 1.04) |
| Heard about FP through the radio last few months | | | | | | | |
| No | | 1.00 | 1.00 | | | 1.00 | 1.00 |
| Yes | | 1.00 (0.96, 1.04) | 0.95 (0.89, 1.01) | | | 1.04 (0.99, 1.08) | 0.95 (0.90, 1.01) |
| Heard family planning in newspaper magazine last few months | | | | | | | |
| No | | 1.00 | 1.00 | | | 1.00 | 1.00 |
| Yes | | 0.84 (0.77, 0.93) | 0.86 (0.76, 0.97) | | | 0.89 (0.80, 0.96)*** | 0.83 (0.73, 0.93)** |

*(Continued)*

**Table 3.** (Continued)

| Characteristics | Null model | Model 1 | | Model 2 | | Model 3 | |
|---|---|---|---|---|---|---|---|
| | | Individual-level characteristics | | Community-level characteristics | | Both individual and community-level characteristics | |
| | | Unmet need for spacing RRR (95%CI) | Unmet need for limiting RRR (95%CI) | Unmet need for spacing RRR (95%CI) | Unmet need for limiting RRR (95%CI) | Unmet need for spacing RRR (95%CI) | Unmet need for limiting RRR (95%CI) |
| Heard FP on television last few months No Yes | | 1.00 0.95 (0.89, 1.01) | 1.00 0.02 (0.94, 1.11) | | | 1.00 0.89 (0.84, 0.95)** | 1.00 1.01 (0.94, 1.10) |
| Parity 0 1 2–4 5 and above | | 1.00 1.86 (1.70, 2.05) 2.21 (2.01, 2.44) 2.97 (2.67, 3.31) | 1.00 1.71 (1.11, 2.64) 6.42 (4.41, 9.34) 15.61 (10.69, 22.81) | | | 1.00 1.85 (1.68, 2.04)*** 2.18 (1.98, 2.40)*** 3.90 (2.60, 3.23)*** | 1.00 1.69 (1.10, 2.61)* 6.37 (4.38, 9.27)*** 15.88 (10.87, 23.19)*** |
| Number of under 5 children None One Two Three and above | | 1.00 2.33 (2.16, 2.52) 3.13 (2.89, 3.38) 3.62 (3.32, 3.95) | 1.00 2.16 (1.99, 2.34) 2.34 (2.16, 2.53) 2.27 (2.08, 2.48) | | | 1.00 2.34 (2.16, 2.52)*** 3.10 (2.86, 3.36)*** 3.50 (2.22, 3.83)*** | 1.00 2.17 (2.00, 2.35)*** 2.36 (2.18, 2.56)*** 2.33 (2.14, 2.55)*** |
| Household size 1–2 3–4 5 and above | | 1.00 0.80 (0.72, 0.89) 0.78 (0.70, 0.88) | 1.00 0.83 (0.69, 1.00) 0.88 (0.74, 1.06) | | | 1.00 0.79 (0.71, 0.88)*** 0.77 (0.68, 0.86)*** | 1.00 0.85 (0.71, 1.02) 0.91 (0.76, 1.09) |
| Having knowledge about modern family planning methods No Yes | | 1.00 1.00 (0.93, 1.07) | 1.00 1.33 (1.19, 1.48) | | | 1.00 1.10 (1.02, 1.89)* | 1.00 1.26 (1.12, 1.41)*** |
| *Community-level characteristics of respondents* | | | | | | | |
| Residence Urban Rural | | | | 1.00 1.04 (0.99, 1.08) | 1.00 1.02 (0.97, 1.08) | 1.00 0.86 (0.82, 0.91)*** | 1.00 0.83 (0.77, 0.89)*** |
| Community-level of women illiteracy Low high | | | | 1.00 0.97 (0.93, 1.02) | 1.00 0.88 (0.83, 0.93) | 1.00 0.94 (0.90, 0.99)* | 1.00 0.87 (0.82, 0.92)*** |
| Community-level media exposure Low high | | | | 1.00 0.97 (0.93, 1.01) | 1.00 0.95 (0.89, 1.01) | 1.00 0.98 (0.94, 1.03) | 1.00 0.95 (0.90, 1.01) |
| Community-level child care burden Low high | | | | 1.00 1.05 (1.01, 1.10) | 1.00 1.00 (0.94, 1.05) | 1.00 0.98 (0.94, 1.03) | 1.00 0.97 (0.92, 1.03) |
| Community poverty level Low High | | | | 1.00 1.01 (0.96, 1.06) | 1.00 0.99 (0.93, 1.05) | 1.00 0.97 (0.93, 1.02) | 1.00 0.97 (0.91, 1.03) |

(*Continued*)

**Table 3.** (Continued)

| Characteristics | Null model | Model 1 | | Model 2 | | Model 3 | |
|---|---|---|---|---|---|---|---|
| | | **Individual-level characteristics** | | **Community-level characteristics** | | **Both individual and community-level characteristics** | |
| | | Unmet need for spacing RRR (95%CI) | Unmet need for limiting RRR (95%CI) | Unmet need for spacing RRR (95%CI) | Unmet need for limiting RRR (95%CI) | Unmet need for spacing RRR (95%CI) | Unmet need for limiting RRR (95%CI) |
| African region | | | | | | | |
| Eastern | | | | 1.00 | 1.00 | 1.00 | 1.00 |
| Western | | | | 1.23 (1.18, 1.29) | 0.85 (0.81, 0.90) | 1.25 (1.19, 1.33)*** | 0.87 (0.82, 0.94)** |
| Central | | | | 1.56 (1.48, 1.65) | 0.86 (0.80, 0.94) | 1.48 (1.39, 1.57)*** | 0.85 (0.77, 0.94)** |
| Southern | | | | 0.42 (0.34, 0.53) | 0.93 (0.78, 1.11) | 0.71 (0.57, 0.89)** | 1.47 (1.21, 1.80)** |

Note

* = p<0.05

** = p<0.01

*** = p<0.001.

95%CI; 1.01, 1.11) times higher risk of having an unmet need for limiting. Regarding health care decision making, making a decision with their husband and by their husband only is associated with lower risks of having an unmet need for limiting. Being hearing about family planning through newspapers or magazines was associated with 17% (RRR = 0.83, 95%CI; 0.73, 0.93) lower risks of having an unmet need for limiting. Women with parity of one, 2–4, and 5 and above had 1.69 (RRR = 1.69, 95%CI;1.10, 2.61), 6.37 (RRR = 1.37, 95%CI;4.38, 9.27), and 15.88 (RRR = 15.88, 95%CI; 10.87, 23.19) times higher risks of having an unmet need for limiting as compared to nulliparous women. Having one, two, and three and above under-five children is associated with 2.17 (RRR = 2.17, 95%CI; 2.00, 2.35), 2.36 (RRR = 2.36, 95%CI; 2.18, 2.56), and 2.33 (RRR = 2.33, 95%CI; 2.14, 2.55) times higher risks of having an unmet need for limiting respectively as compared to those with no under-five children. Women who knew about family planning have 1.26 (RRR = 1.26, 95%CI; 2.14, 2.55) times higher risks of having an unmet need for limiting as compared to their counterparts. Moreover, women from rural areas and communities with higher women illiteracy level had 17% (RRR = 0.83, 95%CI; 0.77, 0.89) and 13% (RRR = 0.87, 95%CI; 0.82, 0.92) lower risks of having an unmet need for limiting. There was also regional variation in the unmet need for limiting (Table 3).

**Random effect analysis.** As shown in Table 4, the ICC in the null model revealed that about 14% of the variability in unmet need for family planning was attributed due to differences between clusters/communities. While the highest PCV in the final model revealed that about 22% of the

**Table 4. The random effect model in assessing the factors associated with unmet need for family planning in sub-Saharan Africa.**

| Parameter | Null model | Model 1 | Model 2 | Model 3 |
|---|---|---|---|---|
| Community level variance | 0.520 | 0.500 | 0.494 | 0.406 |
| Intraclass Correlation Coefficient | 0.140 | 0.132 | 0.130 | 0.110 |
| Proportional Change in Variance | Reference | 0.038 | 0.050 | 0.220 |
| Model fitness | | | | |
| Log-likelihood | -122553 | -110832.3 | -122070.7 | -110512 |
| Akaike's information criteria | 245111.9 | 221802.5 | 244179.4 | 221194.1 |
| Bayesian information criteria | 245142.1 | 222497.9 | 244370.8 | 222050.7 |

variability of unmet need for family planning was explained by both individual and community-level factors. Regarding model fitness, the model with the highest log-likelihood and the lowest Akaike's information criteria, model 3, was the best-fitted model (Table 4).

## Discussion

This study aimed to assess factors associated with unmet need for family planning in sub-Saharan Africa. The study at hand identified both individual and community levels factors that are associated with the unmet need for limiting and spacing.

This study revealed that the unmet need for spacing is decreased as a woman goes from the age group 15–19 to the next consecutive age groups (older age groups). Regarding the unmet need for limiting, it is lower as women go from 15 to 19 to 20–24. However, the unmet need for limiting is higher as the age group gets old than 24. This is in line with studies conducted in Ghana, Ethiopia, and Kenya [27–30]. This is since younger-aged women would prefer short-term family planning methods to space their next births and are more likely to have an unmet need for spacing. Besides, this may be because younger women are less likely to have achieved their fertility goals and therefore, want to space than limit the number of births. However, older aged women may have an ideal family size (may have achieved their fertility goals) and tend to prefer long-term contraceptive methods and in some instances permanent methods of contraception and, therefore, they are more likely to have an unmet need for limiting.

This study revealed a higher unmet need for both limiting and spacing among respondents who had completed primary education and among women from communities with higher women illiteracy level. This is in line with study findings from Ghana and Ethiopia [27,31,32]. This could be because women with only a primary education and those from low-literacy communities may unaware of where contraceptive methods can be found and when to use them. However, in this study, women with higher educational level had a lower risk for both unmet needs for limiting and spacing as compared to those without formal education. This is in line with studies conducted elsewhere [29,32,33]. The likely explanation is that, unlike women with no formal education, educated women do not have deep rooted misconceptions about the benefits of contraception and tend to have strong positive attitudes regarding family planning methods [28]. Meanwhile, women with higher partner's education (primary and secondary education as compared with not having formal education) had a higher tendency to experience the unmet need for limiting but not associated with unmet need for spacing.

The study at hand also revealed that having knowledge about modern family planning methods is associated with both unmet needs for spacing and limiting. That means women who had knowledge about family planning have a higher unmet need for spacing and limiting. This is inconsistent with study findings from Ethiopia [34] and Nigeria [35]. The likely reason is that women who are more knowledgeable about family planning methods are more aware of the potential risks, as a result, they may have an increased unmet need due to fear of side effects. However, this is an unusual finding and it needs further investigation.

Those who heard about family planning through media (television and newspaper/magazine) were less likely to experience an unmet need for family planning. This is in agreement with study findings from Ethiopia [30,32] and Pakistan [33]. The explanation for this could be that obtaining information about family planning methods through various media can assist women in learning about and understanding family planning options that are accessible and available.

In this study, age at marriage had associated with the unmet need to limit and space births. That is women who got married at the age of 18 or older were more likely to have a higher level of unmet need for spacing and limiting. This finding is consistent with other studies conducted elsewhere [29,30,36]. This is because, compared to women who were married before

the age of 18, those who married after the age of 18 are more prone to accept attitudes about birth restrictions and, therefore, more likely to have higher demand and unmet need for family planning [15].

The study at hand revealed that the involvement of the husband in women's health care decisions decreases the unmet need for limiting and spacing. This finding is consistent with a study conducted in Burkina Faso [37], which revealed that husbands' approval of health care such as family planning is a protective factor of unmet needs. This may be because the involvement of a husband in health care decision-making, especially decisions about family planning, helps the woman to have an open discussion with her partner and make an appropriate decisions for their health and this may result in a lower unmet need for family planning.

Women with higher numbers of under-five children are more likely to have unmet needs for both limiting and spacing of births than women with no under-five children. This finding is in line with study results from Kenya [28] and Ethiopia [15,30]. In addition, in this study, women with higher parity are more likely to have an unmet need for both spacing and limiting childbirth. This is because these group of women may attain their ideal family sizes (required numbers of children) and hence are more likely to have high unmet need to limit the birth of additional children and/or to delay the next childbirth.

Moreover, in this study, women from higher numbers of households compared to those with one to two household members had higher risks for unmet need for spacing but it is not associated with unmet need for limiting. This finding is agreed with a study done in Afghanistan [19]. This could be because mothers of big families are preoccupied with caring for their family and are unable to seek health care, including family planning [38,39].

In this study, place of residence had significantly associated with both unmet needs for spacing and limiting. Rural dwellers were less likely to report unmet needs for limiting and spacing. This is consistent with a study finding from Ghana [40] and inconsistent with prior studies conducted elsewhere [31,33]. The possible explanation is that rural women have relatively lower decision-making power about family planning use and could not report contraceptive use levels and have lower reports in unmet needs. However, the author recommends further studies in this regard.

The study has both strengths and limitations. It is useful for policymakers and program planners to create intervention strategies because it is a multi-country analysis based on nationally representative data. In addition, for better parameter estimation, we used a multi-level multinomial logistic regression approach. However, investigating causality between dependent and independent factors is difficult due to the cross-sectional nature of the data. Furthermore, because it is reliant on available factors in the data set, the most relevant explanatory variables may be overlooked.

## Conclusion

In this study, the overall magnitude of unmet need for family planning was high. Both individual and community-level factors were associated with both unmet needs for family planning (unmet need for spacing and limiting). Therefore, it is better to give special attention to those women who are at risk of having a higher unmet need for family planning. Moreover, media campaigns related to family planning will have a great impact to reduce the unmet need for family planning.

## Acknowledgments

My deepest gratitude and appreciation go to the measure DHS program for allowing me to use these survey data sets.

## Author Contributions

**Conceptualization:** Achamyeleh Birhanu Teshale.

**Data curation:** Achamyeleh Birhanu Teshale.

**Formal analysis:** Achamyeleh Birhanu Teshale.

**Investigation:** Achamyeleh Birhanu Teshale.

**Methodology:** Achamyeleh Birhanu Teshale.

**Resources:** Achamyeleh Birhanu Teshale.

**Software:** Achamyeleh Birhanu Teshale.

**Validation:** Achamyeleh Birhanu Teshale.

**Visualization:** Achamyeleh Birhanu Teshale.

**Writing – original draft:** Achamyeleh Birhanu Teshale.

**Writing – review & editing:** Achamyeleh Birhanu Teshale.

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
