## [Decision Letter · Decision Letter 0]

12 Jan 2022

PONE-D-21-28601Factors associated with unmet need for family planning in sub-Saharan Africa: A multinomial multilevel logistic regression analysisPLOS ONE

Dear Dr. Teshale,

Thank you for submitting your manuscript to PLOS ONE. After careful consideration, we feel that it has merit but does not fully meet PLOS ONE’s publication criteria as it currently stands. Therefore, we invite you to submit a revised version of the manuscript that addresses the points raised during the review process.

We look forward to receiving your revised manuscript.

Kind regards,

Kannan Navaneetham, PhD

Academic Editor

PLOS ONE

Journal Requirements:

Reviewers' comments:

Reviewer's Responses to Questions

**Comments to the Author**

1. Is the manuscript technically sound, and do the data support the conclusions?

Reviewer #1: Yes

Reviewer #2: Yes

2. Has the statistical analysis been performed appropriately and rigorously? 

Reviewer #1: Yes

Reviewer #2: Yes

3. Have the authors made all data underlying the findings in their manuscript fully available?

Reviewer #1: Yes

Reviewer #2: Yes

4. Is the manuscript presented in an intelligible fashion and written in standard English?

Reviewer #1: Yes

Reviewer #2: Yes

5. Review Comments to the Author

Reviewer #1: General comment

The author is expected to state whether or not there is used/followed guideline such as Strengthening the Reporting of Observational Studies in Epidemiology (STROBE) or others to conduct this study and write the manuscript. Using such guidelines will enhance the validity of the report.

Specific comments

Comment 1

Abstract (background)- the author wrote the facts (the figure) regarding the unmet need for FP. However, the author did not state the research gap. Therefore, reconsider it and add a sentence that indicates the research gap.

Comment 2

Abstract (conclusion)- the conclusion should be in line with the objective of the study. In this case, the author has two major objectives as a research question (i.e., magnitude and factors associated with unmet need). Therefore, the author has to conclude to those major research questions.

Comment 3

Explanatory variables-the author has indicated that both individual and community-level variables were identified after searching different kinds of literature. However, since the source of data is DHS, incorporating explanatory variables from other sources is impossible as we have to only rely on the variables that can be obtained from this secondary data. Therefore, re-write the sentence as: “Both individual and community level variables were incorporated as explanatory variables after reviewing the DHS data”.

Comment 4

Data management and statistical analysis- the author used so many variables for the multivariable analysis. As a result, the chance of having multicollinear variables will be increase. Therefore, the author has to report information regarding multicollinearity and the used technique (s) to diagnose it.

Comment 5

Ethical consideration- specify the date when the data were requested and accessed from the DHS program.

Comment 6

Discussion- reduce the first paragraph only to the objective of the study. Other points written here seem results. Hence, move and reshuffle it under the result section.

Comment 7

Discussion- the author has put justification for every comparable or contrasting result. This is very appreciable. However, the justification sentences have to be short and precise. Therefore, chop those sentences into short form. Also, put references for some of your justifications.

Comment 8

Conclusion- the author put the findings (i.e., the prevalence of unmet need for FP and its correlates) of the study as a conclusion. However, the conclusion needs to have a public health implication which is not stated in this manuscript. Public health implication means what does the prevalence indicate according to the current standards of service (national/global target, plan, etc.)? Again, does your finding is low or high according to the current standards of service? Therefore, the author has to forward a valid conclusion based on the actual findings.

Reviewer #2: Thank you for the invitation to review this manuscript. I read the manuscript with keen interest partly because the author used current DHS data and included several countries. The author sought to assess the factors associated with unmet need for family planning in sub-Saharan Africa and used a multinomial multilevel logistic regression analysis. The study was conducted among 175,820 women of reproductive age and the overall prevalence of unmet need for family was 23.70% whiles unmet need for spacing and limiting were 15.81% and 7.90% respectively. Find below my comments.

Abstract

• Line 26, the author uses “we” as if they are more than one author on the paper. This should be looked at and corrected.

• Lines 41 and 42, author uses the words “community and regional” interchangeably, but these words can mean different descriptions. I suggest the author stick to the word “community”.

Background

• Line 55, the word “terminate” may not have been appropriately use here given the fact that terminate also denotes “abortion”. The author should kindly replace it with a more appropriate word.

Methods

Explanatory variables

• The author talks about married or in a union in lines 83-85 but switch to age at cohabitation in lines 100-101. I suggest the author should kindly be consistent with the married or in a union throughout the manuscript.

• Does “community level of women illiteracy” in line 106 mean the same as “Community-level of women education” in line 110? The author should kindly be consistent with one term throughout the manuscript.

• Lines 107-109 “Except for residence and African region, all other community-level variables were created by aggregating individual-level variables into cluster/community level variables as follows”.

• Inferring from the above sentence, there is a high chance that some of the individual level variables will be correlated with the community level variables since the community level variables were derived from those individual level variables. For example, wealth index and community poverty level might be correlated, and the rest. Therefore, multicollinearity might have been an issue in the final model (model 3) where all these variables were modeled together. The author should check multicollinearity and report appropriately.

• Line 128, “we” should kindly be replaced.

Results

• There are several grammar errors in the results section

• Table 2, maternal age categorization is erroneous. The author should kindly correct it.

• Table 2 and 3, the variable “health care decision” has a category referred to as “others”. This category was significantly associated with unmet need for spacing in the final model. Could the author explain what is meant by “others”? This may go to the Explanatory variable section under methods.

• Line 153, add the “%” to the 43.48.

• Lines 166-157, “The majority of the respondents had not heard about family planning methods through radio, television, and newspaper/magazine, respectively”. The author should kindly indicate the numbers in the above sentence.

• Lines 158-159, does 93.45% constitute almost all?

• Line 173-174, refer to my earlier in lines 83-85.

• Lines 181-183 looks confusing. The author should kindly report RRR in the same direction if they are in the same sentence. This makes it clearer for readers. For example, primary education and higher education had 1.10 higher risk and 0.88 lower risk respectively or 10% higher risk and 12% lower risks. This may be applied appropriately throughout the results section.

Discussion

• There are few grammar error here

• Lines 272-275, “This might be due to women with primary education did not make a wise decision regarding whether they will use contraceptives. Besides, women with a primary level of education may have a higher desire to postpone childbearing to fulfill their goals”.

• What does the author mean by “did not make a wise decision”? This might have been inappropriate to used.

• Besides, if a woman is 45 years old and her education level primary, how will this person postpone childbearing to fulfill her goals? This is not convincing enough, and I suggest the author think about other concrete explanation to this finding.

• Lines 280-282, the author should kindly reference any study that reports of health professionals having less unmet need for family planning compared to other professionals.

• Lines 302-304, the author should provide a reference for the explanation made.

• Lines 312-318 sounds somehow contradictory. If women with higher number of under 5 years children and women with higher parity are more likely to have an unmet need for spacing and limiting, then how come they are also more likely to use contraceptives as stated in line 317-318?

• Lines 322-323, what about those who argue that mothers of big families rather have support systems such as in-laws and other siblings, and therefore might have more time to do other things including seeking for healthcare?

Conclusion

• Radio, newspaper magazine, and television were the media outlets used by the author in this study. Interestingly, radio which is supposed to be widely accessible in countries involved in this study was not significant for both spacing and limiting unmet need on the final model. What then informed the author’s decision to suggest that media campaign would have a great impact to reduce family planning unmet needs?

6. PLOS authors have the option to publish the peer review history of their article (what does this mean?). If published, this will include your full peer review and any attached files.

Reviewer #1: No

Reviewer #2: **Yes: **Maxwell Tii Kumbeni

---

## [Author Response · Author response to Decision Letter 0]

20 Jan 2022

Date: January 20, 2022

Point by point response 

Title: Factors associated with unmet need for family planning in sub-Saharan Africa: A multinomial multilevel logistic regression analysis

Manuscript number: PONE-D-21-28601

Dear editor and reviewers, thank you for the comments and suggestions you gave for this manuscript. I found all the comments and suggestions very constructive and important for the improvement the manuscript. Here, below, is the point by point response for your comments and suggestions, besides, I have incorporated all these in the revised manuscript.

Reviewer #1

The author is expected to state whether or not there is used/followed guideline such as strengthening the Reporting of Observational Studies in Epidemiology (STROBE) or others to conduct this study and write the manuscript. Using such guidelines will enhance the validity of the report.

Author’s response: Thank you very much. I have followed the STROBE guideline/checklist while writing this manuscript, see the revised manuscript. 

Abstract (background)- the author wrote the facts (the figure) regarding the unmet need for FP. However, the author did not state the research gap. Therefore, reconsider it and add a sentence that indicates the research gap.

Author’s response: Thank you. The research gap is incorporated in the revised manuscript, see the abstract (background).

Abstract (conclusion)- the conclusion should be in line with the objective of the study. In this case, the author has two major objectives as a research question (i.e., magnitude and factors associated with unmet need). Therefore, the author has to conclude to those major research questions.

Author’s response: Thank you. The conclusion is amended, it is based on the objectives, in the revised manuscript based on the comment you gave me. 

Explanatory variables-the author has indicated that both individual and community-level variables were identified after searching different kinds of literature. However, since the source of data is DHS, incorporating explanatory variables from other sources is impossible as we have to only rely on the variables that can be obtained from this secondary data. Therefore, re-write the sentence as: “Both individual and community level variables were incorporated as explanatory variables after reviewing the DHS data”.

Author’s response: Thank you for your comment and recommendation. I have modified the sentence to read “Both individual and community level variables were incorporated as explanatory variables after reviewing the DHS data”.

Data management and statistical analysis- the author used so many variables for the multivariable analysis. As a result, the chance of having multicollinear variables will be increase. Therefore, the author has to report information regarding multicollinearity and the used technique (s) to diagnose it.

Author’s response: Thank you very much for raising such very important issue. I have checked Multicollinearity using VIF test and there was no Multicollinearity. That is the VIF ranges from 1.03 to 2.09 with the mean VIF of 1.45, see the data management and statistical analysis section in the revised manuscript.

Ethical consideration- specify the date when the data were requested and accessed from the DHS program.

Author’s response: Thank you. I have accessed the data set on July 2021 within two days of asking the Measure DHS program at https://dhsprogram.com. 

Discussion- reduce the first paragraph only to the objective of the study. Other points written here seem results. Hence, move and reshuffle it under the result section.

Author’s response: Thank you for the comment. I have reduced the first paragraph the discussion section accordingly.

Discussion- the author has put justification for every comparable or contrasting result. This is very appreciable. However, the justification sentences have to be short and precise. Therefore, chop those sentences into short form. Also, put references for some of your justifications.

Author’s response: Thank you. I have considered the issue in the revised manuscript.

Conclusion- the author put the findings (i.e., the prevalence of unmet need for FP and its correlates) of the study as a conclusion. However, the conclusion needs to have a public health implication which is not stated in this manuscript. Public health implication means what does the prevalence indicate according to the current standards of service (national/global target, plan, etc.)? Again, does your finding is low or high according to the current standards of service? Therefore, the author has to forward a valid conclusion based on the actual findings.

Author’s response: Thank you. I have modified the conclusion accordingly in the revised manuscript 

Reviewer #2: 

Abstract

• Line 26, the author uses “we” as if they are more than one author on the paper. This should be looked at and corrected.

• Lines 41 and 42, author uses the words “community and regional” interchangeably, but these words can mean different descriptions. I suggest the author stick to the word “community”.

Author’s response: Thank you. I have considered your comment in the revised manuscript, see the revised abstract.

Background

• Line 55, the word “terminate” may not have been appropriately use here given the fact that terminate also denotes “abortion”. The author should kindly replace it with a more appropriate word.

Author’s response: I have modified the word to read “stop”

Methods

Explanatory variables

• The author talks about married or in a union in lines 83-85 but switch to age at cohabitation in lines 100-101. I suggest the author should kindly be consistent with the married or in a union throughout the manuscript.

Author’s response: Thank you. I have incorporated married women for the analysis and I rewrite the sentences accordingly in the revised manuscript.

Does “community level of women illiteracy” in line 106 mean the same as “Community-level of women education” in line 110? The author should kindly be consistent with one term throughout the manuscript.

Author’s response: Thank you. Community level of women illiteracy was consistently used in the revised manuscript. 

• Lines 107-109 “Except for residence and African region, all other community-level variables were created by aggregating individual-level variables into cluster/community level variables as follows”.

Inferring from the above sentence, there is a high chance that some of the individual level variables will be correlated with the community level variables since the community level variables were derived from those individual level variables. For example, wealth index and community poverty level might be correlated, and the rest. Therefore, multicollinearity might have been an issue in the final model (model 3) where all these variables were modeled together. The author should check multicollinearity and report appropriately.

Author’s response: Thank you. I have checked Multicollinearity using VIF test and there was no Multicollinearity (the VIF ranges from 1.03 to 2.09 with the mean VIF of 1.45). I have incorporated the information showing this in the revised manuscript (see the data management and analysis section). 

Line 128, “we” should kindly be replaced.

Author’s response: Corrected in the revised manuscript. 

Results

• There are several grammar errors in the results section

Author’s response: I have checked the result section and grammatical errors are corrected 

• Table 2, maternal age categorization is erroneous. The author should kindly correct it.

Author’s response: Amended in the revised manuscript.

• Table 2 and 3, the variable “health care decision” has a category referred to as “others”. This category was significantly associated with unmet need for spacing in the final model. Could the author explain what is meant by “others”? This may go to the Explanatory variable section under methods.

Author’s response: It was to mean a person other than the husband and themselves. In includes categories “someone else” and “other individuals” in the survey data and for this study, I have merged these in to “other”. I have clarified this in table 2, as a footnote. 

• Line 153, add the “%” to the 43.48.

Author’s response: Corrected 

• Lines 166-157, “The majority of the respondents had not heard about family planning methods through radio, television, and newspaper/magazine, respectively”. The author should kindly indicate the numbers in the above sentence.

Author’s response: The sentence is amended accordingly in the revised manuscript.

• Lines 158-159, does 93.45% constitute almost all?

Author’s response: Thank you. This was to indicate the figure/number is very high. But, in the revised manuscript, I have modified the sentence and the word “almost all” is deleted. 

• Line 173-174, refer to my earlier in lines 83-85. 

Author’s response: Corrected 

• Lines 181-183 looks confusing. The author should kindly report RRR in the same direction if they are in the same sentence. This makes it clearer for readers. For example, primary education and higher education had 1.10 higher risk and 0.88 lower risk respectively or 10% higher risk and 12% lower risks. This may be applied appropriately throughout the results section.

Author’s response: Thank you. I have considered your suggestion/recommendation in the revised manuscript.

Discussion

• There are few grammar error here

Author’s response: Thank you. I have go through the overall manuscript and I have corrected grammatical errors. 

• Lines 272-275, “This might be due to women with primary education did not make a wise decision regarding whether they will use contraceptives. Besides, women with a primary level of education may have a higher desire to postpone childbearing to fulfill their goals”.

• What does the author mean by “did not make a wise decision”? This might have been inappropriate to used.

• Besides, if a woman is 45 years old and her education level primary, how will this person postpone childbearing to fulfill her goals? This is not convincing enough, and I suggest the author think about other concrete explanation to this finding.

Author’s response: I have amended the sentence in the revised manuscript.

• Lines 280-282, the author should kindly reference any study that reports of health professionals having less unmet need for family planning compared to other professionals.

Author’s response: I have considered the comment in the revised manuscript, the justification is amended in the revised manuscript.

• Lines 302-304, the author should provide a reference for the explanation made.

Author’s response: Reference is added. 

• Lines 312-318 sounds somehow contradictory. If women with higher number of under 5 years children and women with higher parity are more likely to have an unmet need for spacing and limiting, then how come they are also more likely to use contraceptives as stated in line 317-318?

Author’s response: Thank you. It was to mean more likely to have higher demand for family planning and I have modified the sentence in the revised manuscript.

• Lines 322-323, what about those who argue that mothers of big families rather have support systems such as in-laws and other siblings, and therefore might have more time to do other things including seeking for healthcare?

Author’s response: In Ethiopia, most of the mothers with big family are burdened by routine activities (child care, cooking, etc…). Most of them, carry the whole family. Therefore, they gave health care as secondary. And I have putted references for this justification in the revised manuscript. 

Conclusion

• Radio, newspaper magazine, and television were the media outlets used by the author in this study. Interestingly, radio which is supposed to be widely accessible in countries involved in this study was not significant for both spacing and limiting unmet need on the final model. What then informed the author’s decision to suggest that media campaign would have a great impact to reduce family planning unmet needs?

Author’s response: Thank you. Even though I had expected Listening radio as associated factor for both unmet need for spacing and limiting, it was not the case in this study. However, the two media (radio and newspaper/magazine) were associated with unmet need for family planning and when I said media campaign, in the conclusion, it is to mean television and newspaper/magazine.

---

## [Editor Report · Decision Letter 1]

31 Jan 2022

Factors associated with unmet need for family planning in sub-Saharan Africa: A multilevel multinomial logistic regression analysis

PONE-D-21-28601R1

Dear Dr. Teshale,

We’re pleased to inform you that your manuscript has been judged scientifically suitable for publication and will be formally accepted for publication once it meets all outstanding technical requirements.

Kind regards,

Kannan Navaneetham, PhD

Academic Editor

PLOS ONE
---

## [Editor Report · Acceptance letter]

3 Feb 2022

PONE-D-21-28601R1 

Factors associated with unmet need for family planning in sub-Saharan Africa: A multilevel multinomial logistic regression analysis 

Dear Dr. Teshale:

I'm pleased to inform you that your manuscript has been deemed suitable for publication in PLOS ONE. Congratulations! Your manuscript is now with our production department. 

Kind regards, 

on behalf of

Prof. Kannan Navaneetham 

Academic Editor

PLOS ONE